# Twisting Lids Off with Two Hands

**Toru Lin** *    **Zhao-Heng Yin** *    **Haozhi Qi**    **Pieter Abbeel**    **Jitendra Malik**

University of California, Berkeley

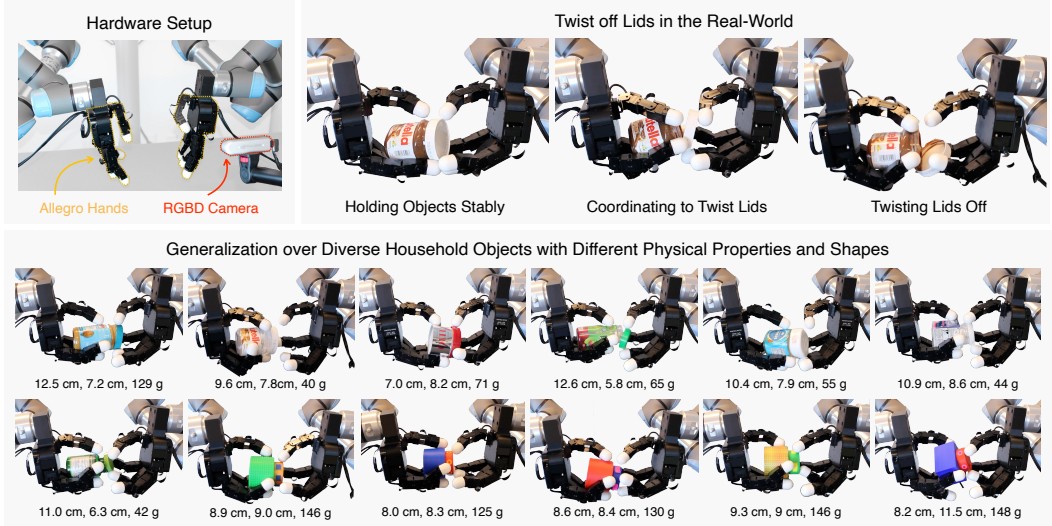

Figure 1: We train two anthropomorphic robot hands to twist (off) lids of various articulated objects. The control policy is first trained in simulation with deep reinforcement learning, then zero-shot transferred to a real-world setup. We show that a single policy trained to manipulate simplistic, simulated bottle-like objects can generalize to real-world objects that have drastically different physical properties (e.g. shape, size, color, material, mass). The length, diameter (or diagonal length), and mass of each object are annotated at the bottom of individual subfigures. More results can be found in our video and and project website.

**Abstract:** Manipulating objects with two multi-fingered hands has been a long-standing challenge in robotics, due to the contact-rich nature of many manipulation tasks and the complexity inherent in coordinating a high-dimensional bimanual system. In this work, we share novel insights into physical modeling, real-time perception, and reward design that enable policies trained in simulation using deep reinforcement learning (RL) to be effectively and efficiently transferred to the real world. Specifically, we consider the problem of twisting lids of various bottle-like objects with two hands, demonstrating policies with generalization capabilities across a diverse set of unseen objects as well as dynamic and dexterous behaviors. To the best of our knowledge, this is the first sim-to-real RL system that enables such capabilities on bimanual multi-fingered hands.

**Keywords:** Bimanual Manipulation, Sim-to-Real, Reinforcement Learning

## 1  Introduction

Achieving dexterous bimanual manipulation with two anthropomorphic robot hands has been exceptionally challenging, due to the inherently contact-rich nature of many manipulation tasks and the complexity of coordinating a high-dimensional bimanual system. This work takes a step towards this grand goal, by demonstrating the feasibility of learning a highly dexterous and dynamic bimanual

---

*Correspondence to {`toru,zhaohengyin`}`@berkeley.edu`. The first two authors contribute equally.

8th Conference on Robot Learning (CoRL 2024), Munich, Germany.

manipulation policy purely in simulation and zero-shot transferring it to the real world. Specifically, we study the task of twisting or removing lids with two multi-fingered robot hands. This task is both practically important and profoundly interesting: for one, the ability to twist or remove lids from containers is a crucial motor skill that toddlers acquire during their early developmental stages [1, 2]; for another, the manipulation skills required for this task, such as the coordination of fingers to manipulate a multi-part object, can be generally useful across a large collection of practical tasks.

Since collecting human expert demonstration data to solve contact-rich tasks via imitation learning is highly challenging and expensive [3], we aim at training a generalizable policy through sim-to-real reinforcement learning without using any expert data (Figure 1). Our method does not require precise modeling of any individual object, or hardcoding prior knowledge on object properties; instead, stable and natural bimanual finger behaviors emerge through large-scale reinforcement learning (RL) training. Below, we share the novel insights that enable us to develop such a system.

**Physical Modeling.** Our work features a novel class of objects for in-hand manipulation: articulated objects defined as two rigid bodies connected via a revolute joint with a threaded structure. Accurately modeling friction and contact with revolute joints and threaded structures has traditionally been a hard challenge in robotic simulation [4]. To address this, we introduce a brake-based design to model the interaction between the lid and body of bottle-like objects. This design is fast to simulate while maintaining high fidelity to real-world physical dynamics, enabling efficient policy learning and successful sim-to-real transfer.

**Perception.** We initially hypothesize that a fine-grained, contact-rich manipulation task like lid-twisting must require precise perceptual information on object states and shapes. To our surprise, a two-point sparse object representation, extracted from off-the-shelf object segmentation and tracking tools, is sufficient to solve the perception problem. With simple domain randomization techniques, we train policies that are robust against occlusion and camera noise. This discovery suggests that a minimal amount of perception information can be adequate for complicated bimanual manipulation tasks.

**Reward Design.** Previously performant reward designs for tasks like in-hand reorientation [5, 6, 7] cannot be straightforwardly applied to our task, since those tasks focus on manipulating single-part rigid bodies with one hand rather than multi-part articulated bodies with two hands. Solving this task is more challenging since it involves more complex and precise contact (e.g. using two hands to hold a lid). In addressing this challenge, we discover a simple keypoint-based contact reward that yields natural lid-twisting behavior on the robot fingers.

We conduct several controlled experiments in both simulation and the real world. Through empirical analysis, we verify that our simulation modeling, perception module, and reward design can reliably lead to the desired behavior of lid-twisting. Our final successful policy manifests natural behavior across test objects with various physical properties such as shapes, sizes, and masses in simulation. Moreover, the learned policy can be zero-shot transferred to a wide range of novel household objects whose lids can be removed (Figure 1), and it is robust against perturbations.

## 2 Background

For decades, bimanual manipulation has remained an unsolved challenge in robotics [8, 9, 10, 11, 12, 13]. While multi-fingered robot hands seem to be a natural choice for bimanual robot systems in theory, designing controllers for high-dimensional action spaces remains an open problem. Classical approach has made significant progress but usually assume known object or physics information [14, 15] and the generalizability remains unknown. In recent years, bimanual manipulation has been actively studied with learning-based methods, as a result of progress in learning algorithms and compute infrastructure. These learning-based approaches can be categorized into two types: 1) learning from real-world data; 2) learning in simulation, then transferring to the real world (sim-to-real).

**Learning from Real-World Data.** Rapid progress has been made in RL in the real world. Zhang et al. [16] learn to chain motor primitives for vegetable cutting, with relatively simple motion primitives; much of the task difficulty is bypassed via the use of specialized end-effectors [17, 18]. Chiu

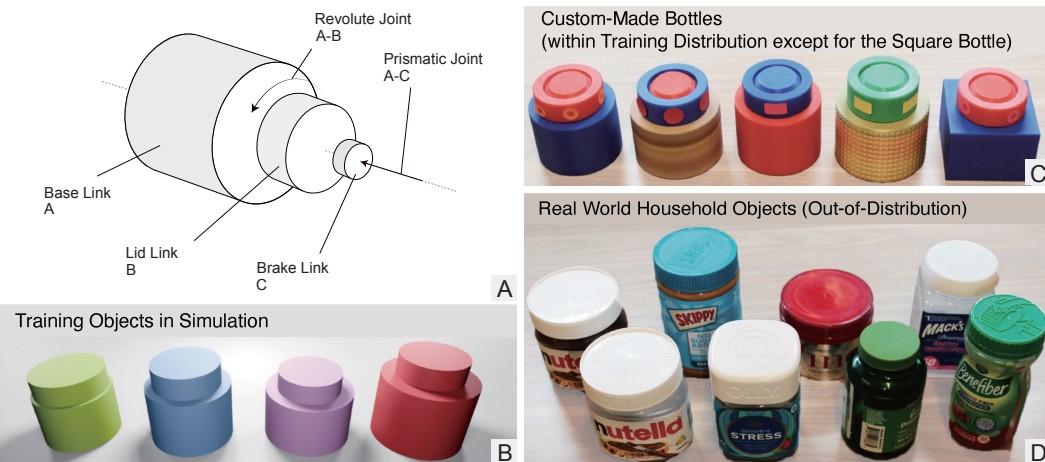

Figure 2: Our bottle model and the used bottles in the simulation and the real world. *A*: Simulated bottle URDF. *B*: Training bottle objects in simulation. *C*: Custom-made bottles (in-distribution except for the rightmost square bottle). *D*: Household object bottles (out-of-distribution).

et al. [19] learn precise needle manipulation with two grippers by integrating RL with a sampling-based planner. While impressive, these works cannot easily scale to higher dimensional action space due to their sample inefficiency or the need to define heuristic-based action primitives.

Most recent successes in bimanual manipulation are achieved by learning from demonstrations [20, 21, 22, 23, 24, 25]. However, successes so far are largely limited to simple end-effectors like parallel jaw grippers due to the lack of high-quality demonstration data from multi-fingered robot hands [26]. Although several works aiming to improve demonstration data collection with two arms [27, 28, 29] or multi-fingered hands [21, 30, 31, 32, 33], their latency and retargeting errors limit their practical applicability and scalability. Lin et al. [20] proposes a scalable hands-arms teleoperation system that learns smooth bimanual policies, but the system compromises on dexterity. Similar systems that offer more dexterous control [34, 35], on the other hand, suffer from drawbacks ranging from jittery control to high costs. Our method uses RL in simulation and is thus not limited by the hardware and data collection infrastructure problems faced by learning from demonstration approaches.

**Sim-to-Real.** There has been growing interest in sim-to-real approaches for robotics – i.e. learning policies in simulation and transferring them to the real world – stimulated by several notable successes in recent years ranging from locomotion [36, 37, 38] to manipulation [5, 7, 6, 39]. Existing works in manipulation, however, are mostly done with either a single multi-fingered hand [3, 40, 41, 42, 43, 44, 45, 46, 47], or two arms with simpler end-effectors [48, 49, 50]. While Chen et al. [51] and Zakka et al. [52] feature bimanual tasks with dexterous hands, only simulation results are shown. The work most related to ours is Huang et al. [53], where the authors demonstrate throwing and catching objects using two dexterous hands. However, our task is significantly more contact-rich and requires substantially more challenging bimanual coordination to maintain object stability at all times. To our best knowledge, there is no learning-based method directly comparable to ours on the proposed task.

## 3 Learning to Twist Lids with Two Hands

We focus on the challenging task of lid-twisting for container objects, since it is a complex in-hand manipulation process that requires dynamic dexterity of multiple fingers and precise coordination between two hands. The goal of this task is to twist the lid about the object's axis of rotation in one direction as much as possible; during this process, the object should always stay in hand. Achieving this involves a sequence of delicate movements: 1) after initialization, the robot hand should firmly grasp and slightly rotate the bottle to a suitable pose; 2) the hand that is closer to the object lid should place its finger around the lid to initiate twisting motion; 3) the two hands should coordinate to avoid dropping the object while one hand twists the lid. Motor skills that arise from this task could serve

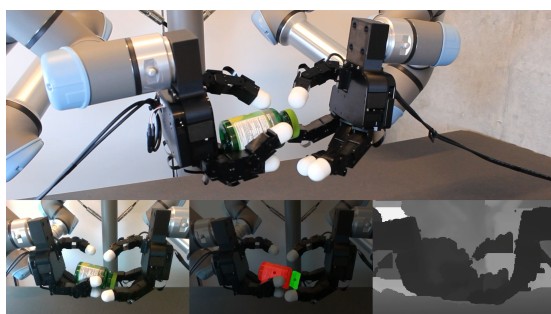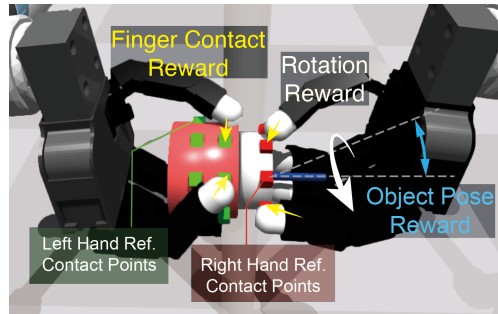

Figure 3: **Left**: Real-time perception system. *Top*: overview. *Bottom*: we segment and track object parts from the RGB frames (left), take mask centers as object part centers (middle), and estimate 3D object keypoints using noisy depth information from the camera (right). **Right**: Illustration of reward design. Our task-specific reward contains finger contact reward (yellow arrows), twisting reward (white arrow), and pose reward (blue arrow). In particular, our keypoint-based finger contact reward is crucial for learning the desired behavior.

as generic abstractions for skills necessary to manipulate many other household objects, especially those with revolute joints such as Rubik's cubes, light bulbs, and jars.

### 3.1 Object Simulation

A central challenge in simulating the lid-twisting task is how to model friction between the bottle body and the lid properly, particularly static friction. Simulating this type of physical force has been a long-standing problem in robotics and graphics [4]. We design a simple modeling approximation that strikes a balance between fidelity and speed during physical simulation; our bottle-like object model is illustrated in Figure 2(A). Our design features a special *Brake Link* that constantly presses against the bottle lid via a prismatic joint. This artificially generates frictional forces between the bottle body (*Base Link*) and the lid (*Lid Link*), preventing relative rotation between them — similar to a bottle with its lid screwed on. We replicate these bottles in the real world, as shown in Figure 2(B).

We note that naively tuning static friction properties between two revolute-joint-connected bodies is not realistic enough with our simulator. Such a brake-based is the only way we find that can simulate the static friction well.

### 3.2 Task Initialization

To better benchmark the bimanual twisting capability, we consider a class of articulated bottles with lids that can be twisted infinitely (see Figure 2(A) and Section 3.1 for more details). Each object of interest consists of two rigid, near-cylindrical parts (a "body" and a "lid"); the two parts are connected via a continuous revolute joint, allowing them to rotate about each other. At the beginning of each episode, the two robotic hands are initialized in a static pose with upward-facing palms, and a bottle-like object is gently dropped or placed onto the fingers. The initial pose of the object is randomized both in translation and rotation to a fixed default pose; the initial joint positions of the hands are randomized about a canonical pose by adding Gaussian noise. Note that since we do not assume a stable grasp configuration at task initialization, the control policy needs to learn in-grasp reorientation to place the object in a stable location to perform successive manipulation.

### 3.3 Policy Learning

Bimanual in-hand dexterous manipulation involves highly complex hand-object contacts, and remains challenging to solve with traditional methods. In this work, we address the control challenge through RL. We formulate our control problem as a partially observable Markov Decision Process.

**Observation Space.** At each time step $t$, the control policy observes the following information from the environment: the proprioceptive hand joint positions $q_t$, the estimated center-of-mass 3D positions of the bottle base and lid, and previously commanded target joint positions $\tilde{q}_t$.

**Action Space.** We use a PD controller to drive the robot hand. The control policy produces a relative target joint position as the action $a_t$, which is added to the current target joint position $q_t$ to produce the next target: $\tilde{q}_{t+1} = \tilde{q}_t + \eta \text{EMA}(a_t)$. $\eta$ is a scaling factor. Note that we smooth the action with

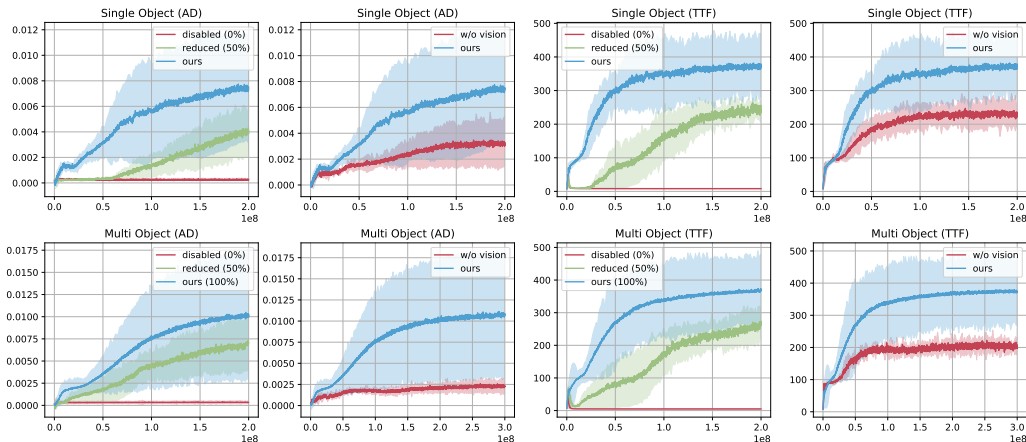

Figure 4: Training curves in different settings. *Top*: Single-object training results (evaluated on single-object setup). *Bottom*: Multi-object training results (evaluated on multi-object setup). *Left half*: Comparisons of different reward setups. *Right half*: Ablations on the use of vision. The results are averaged on 5 seeds. The shaded area shows the standard deviation. The AD score is averaged by the total execution steps.

its exponential moving average (EMA) to produce smooth motion. The next target position is sent to the PD controller to generate torque on each joint.

**Reward.** While one way to approach hard exploration problems is to add intrinsic rewards [54, 55], we introduce the following fine-grained reward terms to shape the hand behavior (Figure 3 right). (1). *Twisting Reward.* We define the twisting reward as $r_{\text{twisting}} = \Delta\theta = q_{bottle}^{t+1} - q_{bottle}^{t}$ which is the rotation angle of the lid during one-step execution. This reward term encourages the hand to twist the lid. (2). *Finger Contact Reward.* We find it crucial to use a set of reference contact points to guide effective contact between fingertips and the bottle. We define two set of points $\mathbf{X}^L \in \mathbb{R}^{n \times 3}$ and $\mathbf{X}^R \in \mathbb{R}^{m \times 3}$ attached on the bottle base and lid respectively. Then, we define the finger contact reward as $r_{\text{contact}} = \sum_i \left[ \frac{1}{1+\alpha d(\mathbf{X}^L, \mathbf{F}_i^L)} + \frac{1}{1+\alpha d(\mathbf{X}^R, \mathbf{F}_i^R)} \right]$, where $\mathbf{F}^L \in \mathbb{R}^{4 \times 3}$ and $\mathbf{F}^R \in \mathbb{R}^{4 \times 3}$ are the position of left and right fingertips, $\alpha$ is a scaling hyperparameter, and $d$ is a distance function defined as $d(\mathbf{A}, \mathbf{x}) = \min_i \|\mathbf{A}_i - \mathbf{x}\|_2$. Therefore, we require each fingertip to stay as close to one of the reference contact points as possible. As we will see later, this term is necessary for eliciting desired behavior and task success. (3). *Pose Reward.* We also introduce a pose matching reward term to encourage the bottle main axis $\mathbf{x}_{axis}$ aligned with a predefined direction $\mathbf{v}$. This term is defined as $r_{\text{pose}} = -\arccos(\langle \mathbf{x}_{axis}, \mathbf{v} \rangle)$. (4). *Regularizations.* Besides the three task-specific rewards, we also introduce another few regularization terms as in previous works [44], including work penalty and action penalty to penalize large, jerky motions. We leave the details of the definition to the appendix.

**Reset Strategy.** There exist many possible hand-object interaction modes. Among these, most modes lead to failures such as getting the object stuck between fingers, and exploring those modes rarely provides good learning signals. To circumvent the high dimensionality of our exploration problem, we introduce two early termination criteria. First, we reset an episode if the robot hands fail to rotate the bottle into a desired pose for bimanual twisting within a short time limit. Additionally, we reset when the bottle's $z$-position is below a certain threshold, as the fingertips of the two hands can pinch the bottle at a low position without being able to reposition it into the palm.

**Training.** We use PPO [56] with asymmetric critic observation [57] to train our policy, and introduce various domain randomizations to make the policy transferable to the real world. We apply both both physical and non-physical randomizations. The detailed training setup can be found in the appendix.

### 3.4 Real-World Perception.

Figure 3 shows an overview of our real-world perception pipeline. To make our RL policy more transferable, we use bottle and lid center points instead of pixels as vision input. We extract these keypoints from real-time images through object segmentation and tracking in the real world. Specif-

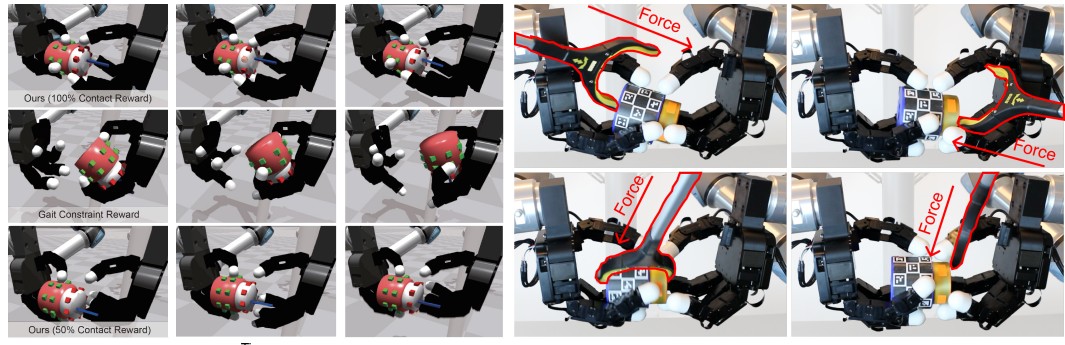

Figure 5: **Left**: Behavior of different reward functions. *Top*: Our full reward function achieves a stable grasp, as well as a smooth, natural, and human-like twisting motion. *Middle*: A naive gait constraint reward without any contact hints leads to erratic finger motion and unnatural grasps. *Bottom*: A reduced contact reward yields somewhat natural behavior, but the grasp is loose compared to the full contact reward case. **Right**: Perturbing a learned policy with random external force. Our policy is resilient to these external forces and able to recover.

ically, we utilize the Segment Anything model [58] to generate two separate masks for the bottle body and the lid on the first frame of each trajectory sequence, and XMem [59] to track the masks throughout all remaining frames. To approximate the 3D center-of-mass coordinates of the bottle body and lid, we calculate the center position of their masks in the image plane, then obtain noisy depth readings from a depth camera to recover a corresponding 3D position. The perception pipeline runs at 10 Hz to match the neural network policy's control frequency.

## 4 Simulation Experiments

To test how to enable the emergence of natural and robust manipulation behaviors, we first conduct several experiments in simulation. Specifically, we study the following questions: How important is the keypoint-based reward for eliciting desired twisting behavior in bimanual manipulation? How important is visual information for solving this task? Is a sparse keypoint representation enough for learning a generalizable policy?

### 4.1 Setup

**Object Set.** In simulated experiments, we use a collection of simulated cylindrical bottles with varying aspect ratios for both training and evaluation. Some samples are visualized in Figure 2(B). We consider two setups in simulation: 1) *multi-object*, in which all the objects are used, and 2) *single-object*, in which a single medium-sized bottle that represents the mean of the dataset is used.

**Evaluation Metric.** We introduce the following metrics for evaluating the performance: 1) *Angular Displacement (AD)* is the total number of degrees through which the lid has been twisted; 2) *Time-to-Fail (TTF)* is the period measured from the moment the bottle is held to the point when it either slips from the hand or becomes lodged; 3) *Velocity (Vel)* is AD divided by TTF, reflecting the speed of twisting motion.

**Baselines.** We compare our policy with the following baselines. 1) *Policy without Vision.* This is a neural network policy without object state information. We use this to evaluate the importance of vision. 2) *Policy with Reduced Contact Reward.* In training this policy, we reduce the intensity of our proposed finger contact reward. We use this to study the role of our contact reward in policy learning and shaping the policy's behavior. 3) *Policy with Gait Constraint Reward.* In training this policy, we replace our contact reward with a gait constraint reward function similar to ones used for in-hand reorientation tasks [39]. This baseline is only used for qualitative analysis.

### 4.2 Results

**Reward Design.** We first compare our approach with the reduced finger reward baseline (Figure 4). After decreasing the scale of finger contact reward, learned policies fail to master the desired lid-twisting skill and have low performance in general. We hypothesize that this is because the

Table 1: Comparison with baselines on real setup. For each method, we deploy 3 policies trained on 3 different seeds and average the results. Each deployment trial is conducted for 30 seconds.

| Method | BlueBottle | | | WoodBottle | | | RedBottle | | | GoldBottle | | | SqaureBottle | | |
|---|---|---|---|---|---|---|---|---|---|---|---|---|---|---|---|
| | AD ↑ | TTF ↑ | Vel ↑ | AD ↑ | TTF ↑ | Vel ↑ | AD ↑ | TTF ↑ | Vel ↑ | AD ↑ | TTF ↑ | Vel ↑ | AD ↑ | TTF ↑ | Vel ↑ |
| Replay | 128.33 ±217.96 | 7.67 ±4.93 | 11.68 ±19.80 | 2.67 ±4.62 | 7.67 ±5.86 | 0.22 ±0.38 | 15.00 ±25.98 | 4.67 ±4.62 | 1.59 ±2.60 | 28.33 ±43.04 | 7.67 ±4.04 | 2.99 ±4.17 | 29.67 ±8.62 | 10.00 ±0.00 | 2.97 ±0.86 |
| No-Vis | 1.33 ±2.31 | 21.67 ±14.43 | 0.04 ±0.08 | 1.07 ±1.85 | 14.67 ±13.61 | 0.27 ±0.46 | 1.90 ±3.29 | 8.33 ±6.11 | 0.27 ±0.47 | 0.67 ±1.15 | 16.33 ±13.05 | 0.04 ±0.08 | 5.00 ±6.24 | 20.33 ±11.24 | 0.18 ±0.20 |
| No-Asym | 18.67 ±28.94 | 30.00 ±0.00 | 0.62 ±0.96 | 0.67 ±1.15 | 19.33 ±15.14 | 0.03 ±0.04 | 8.33 ±14.43 | 13.00 ±15.13 | 0.28 ±0.48 | 4.3 ±7.51 | 3.67 ±3.79 | 0.54 ±0.94 | 0.00 ±0.00 | 2.33 ±1.53 | 0.00 ±0.00 |
| Large | 2.00 ±2.00 | 22.33 ±13.28 | 0.14 ±0.14 | 0.00 ±0.00 | 24.00 ±10.39 | 0.00 ±0.00 | 2.33 ±2.52 | 9.33 ±4.73 | 0.47 ±0.68 | 2.67 ±3.06 | 22.33 ±13.28 | 0.09 ±0.10 | 1.67 ±2.89 | 30.00 ±0.00 | 0.06 ±0.10 |
| Ours | **946.33** ±383.81 | **23.67** ±10.97 | **41.26** ±4.36 | **499.50** ±578.23 | **30.0** ±0.0 | **16.65** ±19.27 | **150.67** ±113.47 | **30.00** ±0.00 | **5.02** ±3.78 | **98.67** ±66.91 | **30.00** ±0.00 | **3.29** ±2.23 | **43.00** ±12.12 | **30.00** ±0.00 | **1.43** ±0.40 |

motion of lid-twisting requires a very specific pose pattern for holding the object; without explicitly encouraging such a pose pattern (e.g., via its contact modes), RL exploration becomes so hard that it is unsolvable within the available training time. We also observe a positive correlation between the intensity of finger contact reward and both 1) sample efficiency during learning and 2) performance of learned policies (as reflected by Figure 4 and qualitative observations in Figure 5 (*left*)).

**Vision vs. No Vision.** We also study the importance of vision modality. Existing works show that certain rotation behaviors can be achieved through implicit tactile sensing (via proprioception) [39]. However, our empirical results show that, in both single and multi-object setups, the no-vision baseline performs substantially worse than our full method. This suggests that knowledge of the position of bottle keypoints is essential for successful lid-twisting.

**Single Object vs Multi Object.** We run RL training with two object settings: 1) using a single bottle-like object; 2) using multiple bottle-like objects with more variation in the ratio between the bottle base and lid. For results shown in Figure 4, all multi-object training runs are evaluated on multi-object setup and all single-object training runs are evaluated on single-object setup. The two settings pose a trade-off between specialization and generalization: in the single-object scenario, the policy might learn successful behaviors more easily but find it harder to generalize to unseen objects, and vice versa. To our surprise, we observe that multi-object training yields slightly better performance compared to single-object training. We hypothesize that multi-object makes exploring lid-twisting behavior an easier process by introducing an object curriculum that covers both easy and hard object instances during training.

## 5 Real-world Experiments

### 5.1 Experiment Setup

**Hardware Setup.** We use two 16-DoF Allegro Hands from Wonik Robotics for our experiments. Each Allegro Hand is mounted on a fixed UR5e arm. We employ a single RealSense D435 depth camera to provide visual information, from which we extract object state information. We send control commands to the robot at a frequency of 10 Hz via a Linux workstation.

**Object Set.** For quantitative evaluation, we evaluate the sim-to-real transfer capabilities of our policies on five different articulated bottle objects (Figure 2(C)). Among them, four are in-distribution (round-body bottles) and one is outside of the training distribution (square-body bottle).

**Evaluation Metric.** We measure both AD and TTF in 20 trials, with each trial lasting for a maximum of 30 seconds. For each evaluated method, we select the three best policies out of ten policies trained on ten different random seeds. We end a trial if the bottle falls off the palm.

**Baselines.** We compare our final policy with the following baselines to study the effect of several key design choices. 1) *Open-loop Replay Policy (Replay).* We record successful trials of our learned policy in the simulation and randomly select a trajectory to replay on the real robot. This baseline is used to evaluate whether the task can be solved by a deterministic motion pattern. 2) *Policy without Vision (No-Vis).* This baseline policy only takes proprioceptive state information as input, without information about the object state. 3) *Policy without Asymmetric Training (No-Asym).* We compare with a baseline where policy is trained without asymmetric PPO, and evaluate whether introducing additional privielged information into the value network will affect the transfer performance. 4)

*Larger Policy Network Size (Large).* We increase the size of our actor-network and train a large-size policy. We use this to evaluate whether over-parameterization harms policy performance.

## 5.2 Twisting Lids in the Real World

We show quantitative results comparing our policy with baseline policies in Table 1. For both metrics, our policy outperforms all baselines across all evaluated objects. Our method can perform stable grasp on all the objects, and can rotate 3 out of 5 objects at a reasonable speed. In particular, for the blue bottle, one of the deployed policies can achieve 4 full turns (360 degrees) in 30 seconds on average. In contrast, almost all the baselines fail to achieve any effective rotations, either getting stuck or dropping the bottle to the ground. We find that the open-loop policy has the lowest TTF score. Replaying a successful trajectory will not lead to a stable grasp for most of the time, and the bottle will directly roll on the fingers and then drop off the palm. This suggests that the considered task involves very fine-grained contacts and requires the policy to act very precisely according to the object state. Another interesting observation is that the large policy does not transfer to the real world, although we confirm that it can achieve similar performance to our full policy in simulation. This suggests that some overfitting occurs, and controlling the size of the policy network is very important for the successful sim-to-real transfer of our considered contact-rich task.

## 5.3 Robustness against Perturbation

Finally, we also evaluate our policy's robustness against force perturbation. Specifically, we perturb the object during deployment at random times by poking or pushing it along random directions using a picker tool (see the right of Figure 5). We find that our policy can reorient and move the object back to a stable pose for continuous manipulation, indicating that it has some robustness against external forces and can adapt to these unexpected changes. Note that we use a marker-based object detection system in this experiment to disentangle the visual occlusion effect.

## 5.4 Exploration of Twisting Lids Off

In the above section, we mainly study whether the twisting behavior can naturally emerge and be transferred to the real world. Next, we explore the limit of our approach by testing it on 10 novel household objects (Figure 2(D)). These objects differ substantially from our training objects in terms of shape, size, mass, material, color, and mechanical design. While the lids of the synthetic bottles that we use for both simulation training and real-world testing can be twisted infinitely, the lids of these household objects cannot. To evaluate our policy's ability to generalize the lid-twisting skill to these novel objects, we use the success rate on a novel yet adjacent *lid-removal* task as the criterion. We define lid-removal as the object's lid being completely detached from the object body (e.g., when the lids fall from the robot's hands in Figure 1.

We find that our policy continues to achieve stable and natural twisting behaviors on these novel objects. Furthermore, while we only train the policy for *lid-twisting*, we also find our policy capable of removing lids. For *HairMask* and *FiberGummies*, our policy showcases lid-removal rates of more than 50%. For particularly challenging objects that require many turns to remove the lid, such as *PeanutButter* and *EmptyNutella*, our policy only achieves 10% lid-removal rates; however, its twisting behavior is steady and robust to perturbation (see our video supplementary for visualization).

## 6 Conclusion

We present an RL-based sim-to-real system for twisting or removing lids of bottle-like objects with two hands. We propose several techniques to handle the challenges that arise: a novel reward design, a sparse object representation for real-time perception, and an efficient yet high-fidelity method to simulate twisting bottle caps. We conduct experiments in both simulation and real world to demonstrate the effectiveness of our approach. Our real-world results show generalization across a wide range of seen and unseen objects.

**Acknowledgments**

We thank Chen Wang and Yuzhe Qin for helpful discussions on hardware setup and simulation of the Allegro Hand. TL is supported by fellowships from the National Science Foundation and UC Berkeley. ZY is supported by funding from InnoHK Centre for Logistics Robotics and ONR MURI N00014-22-1-2773. HQ is supported by the DARPA Machine Common Sense and ONR MURI N00014-21-1-2801.

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

# 7 Object Details

**Simulated Bottles.** We use Isaac Gym [60] to model the simulated learning environments. For the multi-object environment, we use bottles whose bodies range from 82cm to 86cm in diameter and 55cm to 67cm in height, and whose caps range from 62cm to 70cm in diameter and 20cm to 33cm in height. For the single-object environment, we use a bottle whose body is 84cm in diameter and 60cm in height, and whose cap is 67cm in diameter and 26cm in height.

**Real-World Bottles.** We show details of our real-world bottle design in Figure 6.

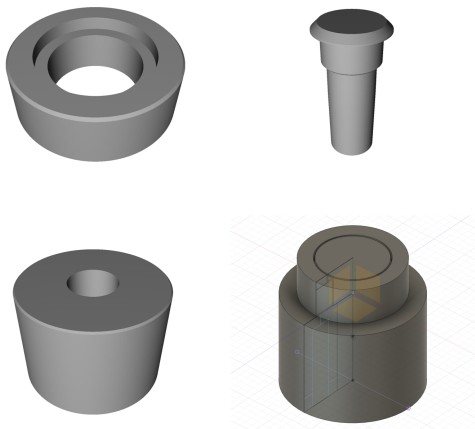

Figure 6: Real-world bottle design. Each bottle is consisted of three parts: the cap (top left), the pin (top right), and the body (bottom left). The parts can be 3D printed and assembled by inserting the pin into the cap and fixing the cap onto the body. With the pin holding the cap and body in place, the cap can be infinitely twisted about the body. Examples of printed bottles are shown in Figure 2(C).

# 8 Real-World Experiment Details

**Camera Calibration.** We use a novel marker-based approach to calibrate the extrinsics matrix of our camera. Specifically, we add the marker tag used in our real-world setup into the simulation environment, such that pair coordinates of the marker corners can be obtained easily in both the camera frame and the world frame (Figure 7). We then use the paired coordinates to solve for camera extrinsics. Doing so greatly reduces the manual labor required by other camera calibration approaches, such as capturing checkerboard images and solving for multiple extrinsic matrices.

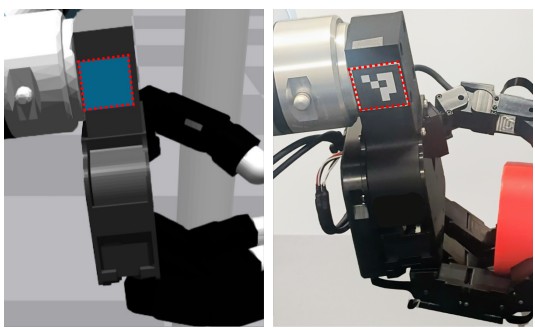

Figure 7: Modeling real-world marker tag in simulation for easy camera calibration.

**Hardware Communication.** To keep our control loop running reliably at 10Hz, we use ZeroMQ to manage communication between robot hands, camera, and Linux workstation.

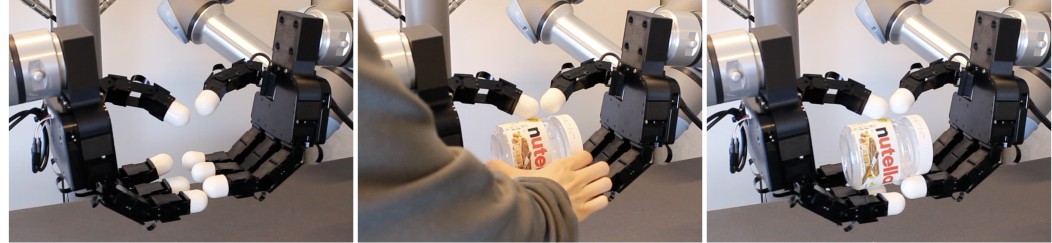

Figure 8: Real-world task initialization. At the beginning of each task sequence, we initialize the robot hands about a canonical position with upward-facing palms. Then, we lightly place an object onto the fingers.

**Task Initialization.** We illustrate details of how we initialize the task sequence in the real world in Figure 8. The canonical joint positions of each finger is documented in Table 2.

Table 2: Initial joint positions of both robot hands.

| Finger | Initial Joint Positions |
|--------|------------------------|
| Index | [-0.0080, 0.9478, 0.6420, -0.0330] |
| Middle | [0.0530, 0.7163, 0.9609, 0.0000] |
| Ring | [0.0000, 0.7811, 0.7868, 0.3454] |
| Thumb | [1.0670, 1.1670, 0.7500, 0.4500] |

# 9 Training Details

**RL implementation.** We use the proximal policy optimization (PPO) algorithm to learn RL policies. We use an advantage clipping coefficient $\epsilon = 0.2$; a horizon length of 16, with $\gamma = 0.99$, and generalized advantage estimator (GAE) [61] coefficient $\tau = 0.95$. The policy network is a three-layer MLP with ELU [62] activation, whose hidden layer is [256, 256, 128]. The policy network outputs a Gaussian distribution with a learnable state-independent standard deviation. The value network is also an MLP with ELU activation, whose hidden layer is [512, 512, 512]. We use an adaptive learning rate with KL threshold of 0.016 [36]. During training, we normalize the state input, value, and advantage. The gradient norm is set to 1.0 and the minibatch size is set to 8192. We use asymmetric observation [57] for the policy and value network, adding privileged information to the value network inputs. This privileged information is not accessible by the policy network.

**Asymmetric States.** In addition to the policy inputs, we provide the following privilege state inputs to the value network of asymmetric PPO: hand joint velocities, all fingertip positions, all contact keypoint positions, object orientation, object velocity, object angular velocity, random forces applied to object, object brake torque, object mass randomization scale, object friction randomization scale, and object shape randomization scale.

**Action Hyperparameters.** To generate action commands, we clip neural network policy output to $[-1, 1]$ range. We then apply an action scale of 0.1 and a moving average parameter of 0.75 to the actions.

**Reward Hyperparameters.** We use $\alpha_1 = 2.5$, $\alpha_2 = 500.0$, $\alpha_3 = 20$, $\alpha_4 = -0.001$, and $\alpha_5 = -1.0$ as reward weights.

**Domain Randomization Setup** We apply a wide range of domain randomizations to ensure zero-shot sim-to-real transfer, including both physical and non-physical randomizations. Physical randomizations include the randomization of object friction, mass, and scale. We also apply random forces to the object to simulate the physical effects that are not implemented by the simulator. Non-physical randomizations model the noise in observation (e.g. joint position measurement and detected object positions) and action. A summary of our randomization attributes and parameters is shown in Table 3.

Table 3: Domain Randomization Setup.

| | |
|---|---|
| Object: Mass (kg) | [0.03, 0.1] |
| Object: Friction | [0.5, 1.5] |
| Object: Shape | $\times\mathcal{U}(0.95, 1.05)$ |
| Object: Initial Position (cm) | $+\mathcal{U}(-0.02, 0.02)$ |
| Object: Initial $z$-orientation | $+\mathcal{U}(-0.75, 0.75)$ |
| Hand: Friction | [0.5, 1.5] |
| PD Controller: P Gain | $\times\mathcal{U}(0.8, 1.1)$ |
| PD Controller: D Gain | $\times\mathcal{U}(0.7, 1.2)$ |
| Random Force: Scale | 2.0 |
| Random Force: Probability | 0.2 |
| Random Force: Decay Coeff. and Interval | 0.99 every 0.1s |
| Bottle Pos Observation: Noise | 0.02 |
| Joint Observation Noise. | $+\mathcal{N}(0, 0.4)$ |
| Action Noise. | $+\mathcal{N}(0, 0.1)$ |
| Frame Lag Probability | 0.1 |
| Action Lag Probability | 0.1 |

## 9.1 Generalization Experiment Details

In Table 4, we provide per-object quantitative results and further analysis on the lid-removal success rate mentioned in Section 5.4. These results show that the overall low lid-removal success rate mostly comes from "hard" objects, i.e. objects that require more turns and/or are more out-of-distribution in shape. We also note that, if we define "number of turns needed to remove lids" of in-distribution objects to be 1, the success rates are consistently 100% for the in-distribution objects.

Table 4: Real-world objects differ greatly in the design of lids. Each object requires a different number of turns to remove the lids, and the difficulty of manipulating different object shapes also varies. In this table, objects are sorted by number of turns needed to remove lids (high to low) and then lid-removal success rate (low to high). Note that lid-removal is a novel task rather than the training task.

| Object | No. of Turns | Lid-Removal % |
|---|---|---|
| PeanutButter | 5 | 10 |
| EmptyNutella | 5 | 10 |
| Nutella | 5 | 40 |
| FiberGummies | 5 | 50 |
| Earplugs | 3 | 20 |
| OilCapsules | 3 | 40 |
| StressGummies | 1 | 40 |
| HairMask | 1 | 60 |
| Overall | - | 33.75 |

## 9.2 Generalization to a Vertical Task Setup

To showcase the generalizability of our approach, we train policies with a novel vertical setup (i.e. the agent opens lids of bottles held vertically). Other than changing the initialization setup (see Figure 9 and Table 5) and turning off the perception system, no change is made to the system components proposed in our work. We additionally note that the horizontal setup in our main text is more challenging than the vertical setup. The vertical setup prevents the most common failure case - lack of stabilization and dropping objects – by design.

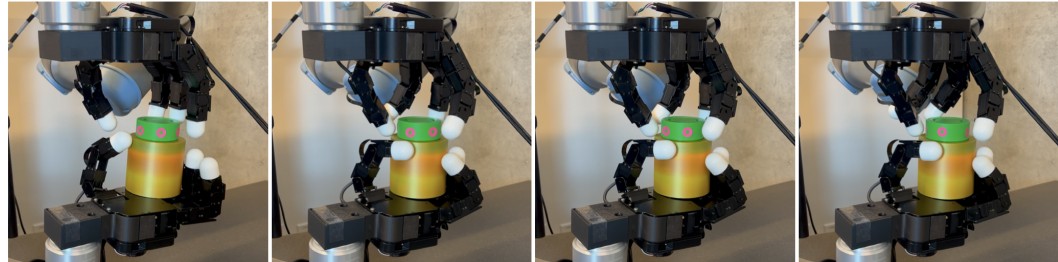

Figure 9: A successful lid-twisting policy in a novel vertical setup. The leftmost image shows task initialization of the vertical setup. The remaining three images show action trajectory of a successful lid-twisting policy being deployed. Additional results can be found in the video supplementary materials.

Table 5: Initial joint positions of robot hands for a vertical task setup.

| Finger | Initial Joint Positions |
|---|---|
| Left: Index | [-0.0080, 0.0772, 1.6655, 0.2697] |
| Left: Middle | [0.0530, 0.0031, 1.7090, 0.0000] |
| Left: Ring | [0.0000, -0.0617, 1.5400, 0.3454] |
| Left: Thumb | [0.6670, 1.1670, 1.0000, 0.8800] |
| Right: Index | [-0.0080, 0.9478, 0.6420, -0.0330] |
| Right: Middle | [0.0530, 0.7163, 0.9609, 0.0000] |
| Right: Ring | [0.0000, 0.7811, 0.7868, 0.3454] |
| Right: Thumb | [0.6670, 1.1670, 0.7500, 0.4500] |

## 9.3 Additional Details on Domain Randomization

During the process of hyperparameter tuning, we note that the highest policy variances are introduced by the following parameters: Bottle Position Observation Noise, Joint Observation Noise, Action Noise. This suggests that noise parameters relevant to action space and observation space might be the most important for domain randomization for successful sim-to-real transfer.

