# OpenReview forum: "Twisting Lids Off with Two Hands"
_robot-learning.org/CoRL/2024/Conference — CoRL 2024_

### Official Review · Reviewer_qU69 · 2024-07-17

**Originality:** 4
**Technical Quality:** 5
**Clarity Of Presentation:** 4
**Potential Impact:** 3
**Recommendation:** 3
**Confidence:** 4

**Review:**

### Strength:

- The paper is well-written, with clear and detailed experimental setups.
- The novel coarse-grained simulation effectively mimics twistable lids by combining revolute and prismatic joints, ingeniously simulating frictional forces.
- The authors achieve zero-shot sim-to-real transfer without requiring an adaptation phase.
- The work demonstrates a robust real-world implementation to problems with variable physical properties using two Allegro Hands.
- The reward engineering is elegant, incorporating a twist reward, finger contact reward, and pose reward, with comprehensive ablation studies to validate their contributions.

### Weakness:

- While the authors successfully demonstrate the bi-manual twist motion, it is not clear from the videos if the motion is sufficient to overcome the resistance at the thread in a fully closed real bottle.
- One of the drawbacks of using a prismatic joint to mimic friction is that it is hard to train the agent to hold the lid after it has come off. That probably explains the videos where the lids fall off to the floor after the twisting motion is complete. However, I acknowledge that is not part of the problem definition.

**Quality Of The Limitations Section:**

3

**Questions For Rebuttal:**

- While the results and the videos are impressive, from the videos, it appears that in many containers (e.g., the Nutella 01:39, StressGummies 01:45), the lid is quite loose even before the twist starts. Maybe one way to mitigate that perception from a casual observer is to simply turn the bottle upside down (e.g., through manual control) before the twisting action starts to show that twisting is essential to open and it wouldn’t open with random jerks of the hands.

- In line 31: you mention “articulated objects defined as two rigid bodies connected via a revolute joint with a threaded structure.” In the Isaac Gym simulation, do you have a threaded structure or do you assume that the prismatic joint A-C in Fig 2A can mimic the interaction of the components at the thread?

- Have you tested (or have arguments against) having a firmer fixed grasp on the base-bottle hand and the RL restricted to the other hand? Wouldn’t that result in a stable outcome, similar to how a human would perform the same manoeuvre?

- Line 60: “Classical …. ”, A word seems to be missing in the beginning.

- Line 175: “To test how to enable natural and robust manipulation behavior emerge”: Grammatical error in line.

**Robotics Focus:**

4

**Summary Of Paper:**

This paper addresses the challenge of learning cooperative behaviour for two robot hands to twist the lid off containers. The authors develop a novel coarse-grain simulation to model the complex interaction between the bottle and lid. Their approach achieves zero-shot transfer, and the learned policies generalise to objects with varying physical attributes that were not seen during simulation without an adaptation phase. The coordination of fingers, a particularly challenging task, is effectively managed through deep reinforcement learning combined with sim-to-real transfer, showcasing dynamic and dexterous behaviours across diverse unseen objects.

**Summary Of Recommendation:**

The paper presents a well-executed study with clear experiments, innovative simulation techniques, successful zero-shot sim-to-real transfer, and robust real-world application validated through thorough ablation studies. I recommend accepting the work.

---

### Official Review · Reviewer_DcnM · 2024-07-20
**Bimanual multi-finger control**

**Originality:** 3
**Technical Quality:** 3
**Clarity Of Presentation:** 4
**Potential Impact:** 3
**Recommendation:** 3
**Confidence:** 4

**Review:**

Strengths:

- the paper is fairly well written and clear in most respects. The problem is well motivated, and the Sim2RL approach seems natural and straightforward

- with increasing interest in humanoid like form factors, bimanual multi-finger control is critical for success. The paper provides an interesting case study in the power of sim2real approaches when paired with simple perception heuristics and reward design.

- zero-shot sim2real transfer is an encouraging finding for biarm dexterity.

Weaknesses

- Algorithmic novelty is somewhat low paired with a fairly narrow task focus

- the rewards are hand-engineered, a textbook RL approach is used. Preliminarily, the value of this work is in demonstrating that high-dof dexterous tasks are still within the reach of current methods provided the physical modeling and sim environment is carefully prepared for learning.

- Task success is measured by soft metrics like lid twisting degrees and time-to-fall. When benchmarked on a harder metric like lid removal rates, the paper reports 50% on seen bottles and 10% on more challenging instances. This would suggest that there is a long way to go.

**Quality Of The Limitations Section:**

3

**Questions For Rebuttal:**

- The observation space seems too impoverished to lead to a robust policy. The bottle+lids are distilled into 3D positions of the center of mass losing significant information about the object shape.  Why not build on point cloud or implicit representations?

- It would seem that lid removal and determining number of twists needed would require some force feedback. If a compliant controller is running under the hood, one may need to adjust the gains based on friction. Why is the lid removal success low and how can it be improved.

- The paper uses highly engineered rewards. Is the performance sensitive to its choice?

- Angular Displacement (AD) - how is this measured accurately in practice on the real objects if there is slippage?

- Baselines feel more like ablations, but there is hardly any comparable work on a task like this. If so, please clarify in related works section.

**Robotics Focus:**

4

**Summary Of Paper:**

Summary: This paper presents a sim2real RL approach to learn bimanual multi-finger control policies for the task of twisting lids of bottles with two hands.

**Summary Of Recommendation:**

Limited novelty, but good contribution that shows sim2real can work well for bi-hand dexterity.

---

### Official Review · Reviewer_qvJe · 2024-07-21
**Good sim to real robot results**

**Originality:** 3
**Technical Quality:** 3
**Clarity Of Presentation:** 3
**Potential Impact:** 3
**Recommendation:** 3
**Confidence:** 4

**Review:**

**Strength**:

1. The paper had good real robot results showing successful sim-to-real transfer
2. Works on many different containers and generalizes to unseen ones
3. For the most part, the paper is up-front about the scope of results which I appreciate. The title rightly says “Twisting Lids (0ff)” instead of just “Twisting Lids” since the policy cannot put a lid on a container or tighten a lid
4. Extensively tested across training seeds - Experiments with 10 seeds, of which the top 3 are evaluated on a real robot

**Weakness**:

I like the results in the paper. All weaknesses mentioned below are in my understanding addressable and are meant to improve the quality of the paper.

Sim experiments need a clearer representation

1. A single table should show scores of ablation, just like Table 1 shows real robot results. Ideally, the same table should compare sim and real scores so we have a clear picture in one place.
2. Fig. 4 is difficult to interpret. AD and TFF are interleaved and it’s difficult to go back and forth between the figures and understand what worked best. Could we have multiple line plots on the same figure so it’s easier to compare? Also feel TFF plots are not necessary. Removing them will reduce confusion and create space for the table mentioned above.
3. Line 213 says multi-object makes exploring easier. Does it improve the score even if the evaluation is on a single object? Both Fig. 4 and the text should make it clear what the training and evaluations are on. For a fair comparison, for multi-object training, single-object evaluation scores need to be presented.

Evaluation criteria:

1. TFF (Time-to-Fail): A higher value is claimed to be better. However, a slow policy could do the same or less no. of turns and yet achieve a higher TFF. Table 1 shows such an example where for the Blue Bottle, No-Asym achieves lower AD but higher TFF than the proposed approach. Yet, the score of the proposed approach is highlighted since AD is higher. I don’t see the value in the TFF metric given its limitations. I think a better metric would be AD/TFF which will indicate how fast the rotations are.

Novelty claim:

1. Line 267 claims twisting lids off to be a novel task/criterion which I don’t feel is right. The task is still to turn the lid, which falls off once it’s turned. Is there a strong reason to call it a novel?

Miscellaneous:
1. Line 237 is confusing “Larger Policy Network Size (Large). We increase the size of our actor-network and train a reduced size policy”. Actor network and policy are the same thing. Also, how much larger is it?
2. There exist teleoperation systems for smooth bimanual operation for anthropomorphic hands, e.g. teleoperation on shadow hand https://youtu.be/J-3CcyF4Emw?si=HZ1mkT79rM1pDutB. I think this should be referenced around line 77
3. In Fig. 1 it would be good to add the simulated scenario as well
4. Typos in line 59. Should it be “Classical methods” instead of “Classical”?

**Quality Of The Limitations Section:**

3

**Questions For Rebuttal:**

Apart from the weaknesses highlighted above,

1. How was the inference delay discrepancy between sim and real (due to additional state estimation, communication delay, etc) handled? It is known to significantly impact sim-to-real performance https://arxiv.org/pdf/2301.12587
2. In line 214, “We hypothesize that multi object makes exploring lid-twisting behavior an easier process by introducing an object curriculum 216 that covers both easy and hard object instances during training.” This hypothesis can be easily validated by (1) evaluating intermediate ckpts of multi-object training on objects of varying difficulty (2) training only on the easiest object
3. In line 143, what are the values of m and n. Is there a special reason to only measure the distance to the nearest point from a set instead of anywhere around a circumference?
4. What do the authors think would take to train a policy that can indefinitely twist a lid?

**Robotics Focus:**

4

**Summary Of Paper:**

This paper presents a method for training a bi-manual anthropomorphic robot to twist lids off containers. The authors train a PPO RL policy using asymmetric actor-critic in simulation. To model the contact dynamics stably, particularly static friction, they use a simplified physical model without threads but with a novel link pushing the lid against the container. In addition, they rely on simple off-the-shelf segmentation models to estimate the noisy positions of the container and the lid. They use reward shaping and domain randomization. The authors show that their simplified model and training techniques successfully transfer to the real world and generalize to unseen geometries. The same policy network can twist lids off different containers with varying degrees of success. For seen container lids, up to 946 degrees (~2.6 turns) and an unseen square bottle’s lid by 43 degrees (refer to table 1). Their policy can remove lids off containers with a 10-60% success rate depending on difficulty, e.g. no. of turns required to remove the lid (refer to table 4 in the appendix).

**Summary Of Recommendation:**

I think the paper has good convincing results but needs some analyses and clearer presentation of results

---

### Author Rebuttal · Authors · 2024-08-12

We thank all reviewers for the helpful feedback. Below, we attach an updated version of our manuscript.

---

### Decision · Program_Chairs · 2024-09-04

**Decision:**

Accept

**Comment:**

**Pre-rebuttal**

This paper receives three positive reviews. Overall the reviewers appreciate the clarity of the presentation and the impressive zero-shot sim-to-real transfer.

Mentioned strengths:
- Relevance to humanoid (**DcnM**).
- Innovation of the simulation technique for the task (**qU69**).
- Thoroughness of ablation studies and model evaluation (**qU69**, **qvJe**).

Mentioned weaknesses:
- Limited algorithmic novelty (**DcnM**).
- Soundness of the TFF metric (**DcnM**, **qvJe**).
- Task remains largely unsolved (i.e. <10% removal rate on unseen objects) even with a fairly narrow task focus (**DcnM**).
- Lid is always lose at the start; unclear whether it can work on fully closed real bottle (**qU69**).
- Presentation of experiments and results (**qvJe**).

---
**Post-rebuttal**

This paper initially received three weak accepts. All the reviewers remain positive after the rebuttal.

Overall the paper has delivered a solid simulation and training framework for an interesting task and demonstrated impressive sim-to-real transfer. AC agrees with the assessments of the reviewers and thereby recommends accept.